# In smiles we trust? Smiling in the context of antisocial and borderline personality pathology

**Lawrence Ian Reed** [1,2]* , **Ashley K. Meyer** [3] , **Sara J. Okun** [1] , **Cheryl K. Best** [4] , **Jill M. Hooley** [4]

**1** Department of Psychology, New York University, New York, New York, United States of America, **2** Department of Psychiatry, McLean Hospital, Harvard Medical School, Belmont, Massachusetts, United States of America, **3** Department of Psychiatry, Depression Clinical and Research Program (DCRP), Massachusetts General Hospital, Boston, Massachusetts, United States of America, **4** Department of Psychology, Harvard University, Cambridge, Massachusetts, United States of America

☯ These authors contributed equally to this work.
* lr113@nyu.edu

**Data Availability Statement:** Our data are publicly available here: https://osf.io/yzrvs/?view_only=8878cda15a6242599fd3fbde8de40454.

**Funding:** The authors received no specific funding for this work.

## Abstract

Research suggests that people behave more cooperatively towards those who smile and less cooperatively towards those with personality pathology. Here, we integrated these two lines of research to model the combined effects of smiles and personality pathology on trust. In two experiments, participants read vignettes portraying a person with either borderline personality disorder, antisocial personality disorder, or no personality pathology. These portrayals were paired with a brief video clip that showed a person with either a neutral expression or a smile. Participants then played a Trust game with the "person" presented using each vignette and video clip combination. In Experiment 1, rates of trust were lower in response to the borderline and antisocial personality disorder vignettes compared with the control vignette. Interestingly, the effect of smiles was dependent upon personality. Although participants were more trusting of smiling confederates portrayed as having borderline personality disorder or no pathology, they were less trusting of confederates portrayed as having antisocial personality disorder if they smiled. In Experiment 2, run with a second set of personality vignettes, rates of trust were lower in response to both personality disorder vignettes and higher in response to smiles with no significant interaction. Together, these results suggest that information regarding both the current emotional state as well as the personality traits of a partner are important for creating trust.

## In smiles we trust? The effects of antisocial and borderline personality pathology

Many of the most consequential decisions we make involve whom to trust. If a partner is trustworthy, prosocial behavior could result in the benefits inherent in cooperation and/or the division of labor. However, if a partner is not trustworthy, prosocial behavior could result in a

**Competing interests:** The authors have declared that no competing interests exist.

damaging loss. As such, any information that would increase our ability to judge trustworthiness would be of value. Independent lines of research on personality pathology and facial expression suggest that both might play a role.

The role of personality pathology has been examined in two recent studies measuring behavioral responses of participants playing economic games with partners portrayed using vignettes describing individuals with DSM-5 personality disorders. Studying cooperation, Reed, Best, and Hooley [1] had participants play cooperative games (either a one-shot Prisoner's Dilemma or Chicken game) with a confederate portrayed using one of eight personality disorder vignettes (antisocial, avoidant, borderline, dependent, histrionic, narcissistic, schizoid or schizotypal) or a control vignette. In the Prisoner's Dilemma game, participants were less likely to cooperate with confederates portrayed using each of the personality disorder vignettes in comparison to the control vignette. Similar results were found in the less competitive Chicken game, where participants were less likely to cooperate with confederates portrayed using all but one of the personality disorder vignettes (schizoid personality disorder) in comparison to the control vignette.

In a follow-up study on bargaining, participants played an Ultimatum game (as the proposer) or a Dictator game (as the allocator) with confederates portrayed using the same set of vignettes [2]. In the Ultimatum game, participants offered less money to confederates portrayed through the antisocial, avoidant, dependent, histrionic, narcissistic, and schizotypal vignettes in comparison to the control vignette. In the Dictator game, participants allocated less money to confederates portrayed using the antisocial, borderline, dependent, histrionic, narcissistic, and schizoid vignettes in comparison to the control vignette. Taken together, results from these studies suggest people are less cooperative, make more competitive bargains, and are less giving towards individuals with personality pathology.

Results from several studies examining the behaviors of individuals with personality pathology suggest that such interactive approaches may not be entirely unwarranted. When participants with borderline personality disorder are asked to play the role of the investor they give smaller investments in a one-shot Trust game [3] and are less able to maintain and repair cooperation in an iterated Trust game [4] in comparison to controls. Similar results have been found among those with psychopathy in the Prisoner's Dilemma game. Rilling and colleagues [5] found that participants scoring higher in psychopathy were more likely to defect over time in an iterated game. In addition, Mokros and colleagues [6] found that criminal psychopaths were 7 times less likely to cooperate in comparison to adults from the general population.

Whereas knowledge of a partner's personality pathology inhibits prosocial behavior, studies of facial expression show that smiles promote prosocial behavior. Those who smile are judged to be more altruistic and sociable than those who do not [7]. Furthermore, these judgements seem to affect behavior when people interact with those who smile. Smiles have been shown to elicit greater allocations in a Dictator game [8], greater investments in a Trust game [9], greater rates of cooperation in a Prisoner's Dilemma game [10], and more credibility in a credibility assessment task [11]. Like the judgements others make of those with personality pathology described above, the judgements made of those who smile are often accurate. Smiling individuals have been found to score higher on an altruism scale in comparison to those who do not [12–14]. Finally, smiling participants are more likely to share [12] and more likely to cooperate in a Prisoner's Dilemma game [10] than those who do not.

Integrating these two lines of research, the current study investigates the combined effects of personality pathology and smiles on trust. We aimed to examine trust to further extend our previous research on personality pathology, facial expression, cooperation, and bargaining. Previous work has demonstrated that smiles increase prosocial behavior [10]. The presence of borderline and antisocial personality pathology also influences cooperation and bargaining

behavior in those who play economic games with them [1, 2]. Here, we examined participants' behavior in a one-shot Trust game towards confederates who varied in both personality pathology (using vignettes describing borderline personality disorder, antisocial personality disorder, or no pathology) and facial expression (displaying either a neutral expression or a smile). We chose the one-shot trust game [15] as it has been found to be a valid measure of trust [16, 17].

Both borderline and antisocial personality disorders are defined polythetically (i.e. a person must meet a minimum number of diagnostic criteria to warrant a diagnosis) [18]. As such, there exists a great deal of potential heterogeneity among individuals who are diagnosed with either form of personality disorder. To examine the potential effects of varying the presentation of personality pathology, we modified the personality disorder vignettes in a second experiment to portray another phenotype with several defining features of each personality disorder functioning in a community setting.

Based on previous research, we hypothesized that confederate trustees whose descriptions contained features of either borderline or antisocial personality disorder would be judged as less trustworthy and given smaller investments from participant investors in comparison to those described as having no pathology. We also hypothesized that confederates who smiled, regardless of the presence or absence of a personality disorder, would be judged as more trustworthy and given larger investments in comparison to those who displayed a neutral expression. Finally, we examined the combined effects of personality and smiles to explore potential interactions.

## Experiment 1

### Materials and methods

**Participants.** Two hundred and sixty-two participants (109 male, 152 female, and 1 identifying as other) were recruited using Amazon's Mechanical Turk (MTurk), a crowdsourcing web service that coordinates the supply and demand of human interaction tasks (HITS). Sample size was determined before any data analyses. MTurk has been used in previous research in psychology and provides a supportive infrastructure for participant recruitment, screening, payment, and cultural diversity [19, 20]. Participants' self-reported age ranges were between: 18–24 (13.0%), 25–34 (36.6%), 35–44 (23.7%), 45–54 (15.3%), 55–64 (7.3%), and 65–74 (4.2%). Participant's self-reported races were Caucasian (75.6%), African-American (9.2%), Asian (8.4%), and other (6.9%).

**Trust game.** Participants played the role of the investor playing with a confederate trustee in a variant of the Trust game [15, 21]. In this game, the investor begins with a sum of money (say 50 cents). This can either be kept or invested in the trustee. If the investor keeps the money, the game ends with the investor earning 50 cents and the trustee earning nothing. The game continues if the investor invests in the trustee. In this case, the money invested is tripled and the trustee then chooses to either retain or split the money with the investor. If the trustee retains the money, the trustee keeps it and the investor earns nothing. If the trustee splits the money, both players earn half of the invested amount.

In the Trust game, trust is defined as a wager that the trustee will behave reciprocally and split the tripled sum. Trustworthiness is defined as a split of the tripled sum. Trust is risky because a selfish trustee would be expected to keep the entire investment for themselves. As such, an investor anticipating a selfish trustee might invest less (or nothing at all) whereas an investor anticipating a cooperative trustee might invest more.

The game was described to participants as follows:

In this study, you and another person have the opportunity to make money by playing a simple game. In this game, you will choose how much money to share with a partner. The

amount of money that you choose to share will triple, and then your partner will either a) share what they have received with you or b) keep all of it. Based on your and your partner's decisions, the money will be distributed accordingly.

We have met with your partner, interviewed them, and asked them to make a decision about whether they would share or keep the money in this situation. Specifically, they were told that you had the option to share a sum of money with them, but that you could also choose to share none of it, and keep it entirely for yourself, leaving them with no money.

They were made aware that the more you shared with them, the more both of you could make, but they also knew that they could of course keep all the money you sent them, returning none of it to you.

They were then given the choice to either share with you whatever tripled amount they received or keep the entire tripled amount for themselves. We collected their decision and they will be your partner today in this interaction.

To give you a better sense of whom you're interacting with, you'll be asked to read a brief description of them and view a brief video clip of them before you make your decision.

## Game instructions

You will decide how much money you would like to share with your partner. You can choose to share up to 50 cents with your partner. The money that you decide to share will be tripled. For example, if you choose to share 10 cents, it will turn into 30 cents. If you choose to share 40 cents, it will turn into 120 cents.

At that point, based on what your partner told us earlier when we collected their response, they will either share that money with you (allowing you to make more money than you would have if you had kept it all), or they will keep whatever money you sent.

After reading the instructions, participants were required to correctly answer three comprehension questions before continuing with the experiment (e.g., "If you share nothing with your partner, how much money will each player have (in cents)?" and "If you share 10 cents with your partner and they decided not to share with you, how much money will each player have (in cents)?"). Each comprehension question had two required responses, one for the participant and the other for the confederate trustee. Participants were given as many opportunities as necessary to correctly answer these questions.

**Vignettes.** Participants were then randomly presented with a single vignette purportedly describing the trustee paired with a video clip displaying their facial expression. The use of the personality disorder vignettes was similar to that in Reed, Best and Hooley [1] and Reed, Best, Harrison, and Hooley [2]. Briefly, participants were told they would be partnered with a real person who had been interviewed and was being portrayed by each vignette. The antisocial vignette, adapted from Race and Furnham [22] and the borderline vignette, adapted from Millon, Millon, Meagher, Grossman, and Ramnath [23] were intended to portray an individual with several defining features of that personality disorder functioning outside of a hospital setting. The vignette portraying borderline personality disorder read as follows:

Most people think that Kaci, who is 24 years old, lives a life analogous to a soap opera. She is often wrought with emotional ups and downs and is known to be unstable and frequently angry. What fuels the chaos are intense interpersonal needs and sudden shifts of opinion

about others, who may be regarded as loving, sensitive, and intelligent one minute and accused of neglect and betrayal the next. When she is alone, even for a short time, Kaci feels intolerably lonely and empty. Her past relationships have typically been stormy and intense and she spends a lot of her time either making up or breaking up. Kaci often makes frantic attempts to avoid feeling abandoned; on several occasions she has made superficial cuts to her wrists. Kaci lacks a mature sense of self-identity. She often flip-flops on goals and values, suddenly changing jobs on impulse and reversing previous opinions with indifference.

The vignette portraying antisocial personality disorder read as follows:

Tammy is 19 years old and lives with her single mother. Her parents have been divorced for 15 years and her mother finds disciplining her without a father figure quite difficult. She is disobedient and resentful of authority. She is also unwilling to take part in family activities and is violently argumentative when confronted by her mother about her all-night partying. She has been arrested twice for shoplifting and once for driving while intoxicated. Her mother believes that she is doing fairly well at school and is the star player of her basketball team, but she has been lying to her—she never completed high school and was never on the basketball team. Her lying began when she was 12 years old. She was frequently truanting from school and would spend her time loitering in pool clubs smoking cigarettes or in the outskirts of town setting fire to people's property.

We created the control vignette, which was intended to portray an individual without features of any DSM-5 disorder and read as follows:

Jen is 25 and has been married to her husband John for 2 years. They have two children together, Luke who is 2 years old and Penny who is 1 years old. They live in a suburb right outside of town fairly close to where they both grew up. Jen went to a small liberal arts college and majored in psychology. She works as a retail salesperson. In her free time, she likes to play tennis and golf. She played both in high school, but never tried out for the teams while she was in college. She's always had an interest in photography and likes to spend time with her friends hiking, biking, and playing tennis.

**Facial expression stimuli.** The facial expression stimuli were presented directly above the personality vignette. The use of these stimuli was similar to that in Reed, Stratton and Rambeas [11]. Briefly, each clip was 6 s in length and was recorded at 30 frames per second in full color at a resolution of 1260 x 1080 pixels. For each expression clip, we instructed the actress to use facial actions described in the Facial Action Coding System (FACS) [24]. The FACS is a comprehensive, anatomically based system for describing and measuring facial movement. The FACS allows for the creation and coding of facial muscle configurations as combinations of individual Action Units (AUs) [25, 26], providing an objective and reliable description of facial behavior. In the neutral clip, the actress did not produce any expression. The smile clip consisted of the simultaneous action of AU6, cheek raiser; AU12, lip corner puller; and AU25, lips part. Individual AUs were coded independently by a certified FACS coder (L.I.R.). Comparison codes of another FACS coder were used to quantify $\kappa$, which corrects for chance agreement [27]. Agreement between the two coders was high ($\kappa = .92$).

After reading the vignette and viewing the facial expression stimuli, participants were required to correctly answer a single, multiple-choice comprehension question to ensure that participants paid close attention to the vignettes. Participants then specified the amount

they chose to share with the confederate trustee (between 0 and 50 cents). Although participants were told they would be paired with a real person (portrayed using the vignette and video clip), they were never actually paired with anyone. Each participant was paid the highest amount given their decision in response to the vignette. After sharing with the confederate trustee, participants were shown the clip and vignette again and asked to give ratings of the confederate's happiness using a 1–7 Likert-type scale. All participants then answered demographic questions regarding sex, age, and race. Finally, all participants viewed a debriefing statement explaining the interaction, the need for deception, and use of personality disorder vignettes. Participants were paid $2.00 for completing the study. Importantly, participants were told that the monetary payoffs they earned in the game would be paid to them (via MTurk bonus payments) before engaging in the task. Participants completed the task in approximately 7 to 10 min. All procedures were approved by the New York University IRB.

## Results

**Preliminary results.** As a manipulation check, we first conducted a 3 (personality) x 2 (expression) analysis of variance (ANOVA) to examine the effects on participants' ratings of the confederate's perceived happiness. There was a significant main effect for personality, $F(2, 256) = 4.34$, $p = .014$, η2 = .033. Post hoc LSD tests revealed no significant difference between participants who read the control vignette and those who read the borderline vignette, mean difference = .25 ($SE$ = .21), $p = .223$, 95% confidence interval (CI) = [-.15, .66]. However, participants who read the antisocial vignette rated the person depicted in the vignette as significantly less happy than participants who read the control vignette (mean difference = .59; $SE$ = .21; $p = .005$, 95% CI = [.18, .99]). Ratings of perceived happiness did not differ significantly between participants who read the borderline vignette and those who read the antisocial vignette, mean difference = .33 ($SE$ = .21), $p = .109$, 95% CI = [-.07, .74].

There was also a significant main effect for expression, $F(1, 256) = 386.41$, $p < .001$, η2 = .601. Averaged across personality vignettes, perceived happiness ratings were higher among those who viewed the smile ($M = 5.41$, $SD = 1.45$) compared to those who viewed the neutral expression ($M = 2.09$, $SE = 1.30$). There was no significant personality by expression interaction, $F(1, 256) = .0.28$, $p = .755$, η2 = .002.

**Primary results.** A 3 (personality) x 2 (expression) ANOVA was conducted to examine the effects of personality and facial expression on participants' investments. There was a significant main effect of personality, $F(2, 256) = 11.67$, $p < .001$, η2 = .084, but no significant main effect of expression, $F(1, 256) = 0.73$, $p = .394$, η2 = .003. These main effects were also qualified by a significant interaction, $F(2, 256) = 3.82$, $p = .023$, η2 = .029.

The significant personality by expression interaction effect was analyzed using a simple main effects analysis. Among participants who viewed the neutral expression, investments did not significantly differ across personality vignettes $F(2, 256) = 2.843$, $p = .060$, η2 = .022, although investments in partners depicted with features of borderline or antisocial personality were lower (21.44 cents; SD = 17.66 and 23.66 cents; SD = 21.37 respectively) than they were for the control vignette (30.47 cents; SD = 18.65). However, if participants viewed the smile, investments differed significantly across the personality vignettes, $F(2, 256) = 12.967$, $p < .001$, η2 = .092. Post-hoc LSD tests revealed that, after having viewed a smile, participants who read the control vignette invested more (36.39 cents; SD = 16.7) than participants who read the borderline vignette (28.30 cents; SD = 16.35), mean difference = 7.99, $SE$ = 3.87; $p = .040$, 95% confidence interval (CI) for the mean difference = [0.37, 15.62]. They also invested significantly more (mean difference = 19.51; $SE$ = 3.85; $p < .001$, 95% CI for the mean difference =

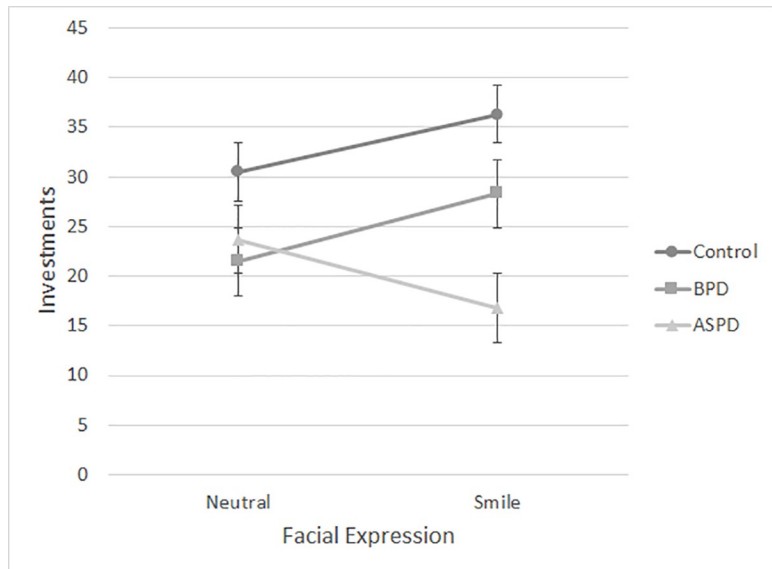

**Fig 1. Money invested (in cents) by vignette and facial expression, Experiment 1.** BPD = borderline personality disorder. ASPD = antisocial personality disorder.

[11.93, 27.30]) after reading the control vignette than they did after reading the antisocial vignette where the mean investment was only 16.78 cents (SD = 19.07). There was also a significant mean difference of 11.52 (*SE* = 3.87) between the investments made by participants who read the borderline vignette in comparison to those who read the antisocial vignette, *p* = .003, 95% (CI) for the mean difference = [3.89, 19.15]. After having viewed a smile, participants gave less money to the person depicted in the antisocial vignette compared to the person depicted in the borderline vignette. The data of interest are presented in Fig 1.

Results from Experiment 1 suggest that both knowledge of a partner's personality pathology and facial expression affect trust. This is consistent with previous behavioral work examining personality pathology [1, 2] and smiles [7–10] in economic games. In response to the borderline personality disorder vignette, these effects were independent. However, in response to the antisocial personality disorder vignette, the effects combined to create an antagonistic interaction.

## Experiment 2

In Experiment 2, we modified the personality disorder vignettes to portray another phenotype with several defining features of each personality disorder functioning in a community (non-hospital) setting. As in Experiment 1, we hypothesized that confederate trustees described as having personality pathology would be judged to be less trustworthy and given smaller investments from participant investors in comparison to those described as having no pathology. We also hypothesized that confederates who smiled would be judged as more trustworthy and given larger investments in comparison to those who displayed a neutral expression. Also, as in Experiment 1, we examined the combined effects of personality and smiles to explore potential interactions.

### Materials and methods

**Participants.** An additional two hundred and eighty-three participants (167 male, 114 female and 2 identifying as other) were recruited using MTurk. Participants' self-reported age

ranges were between: 18–24 (7.8%), 25–34 (49.5%), 35–44 (26.1%), 45–54 (10.2%), 55–64 (4.6%), and 65–74 (1.8%). Participant's self-reported races were Caucasian (79.9%), African-American (10.2%), Asian (6.7%), and other (3.2%).

**Trust game.**    Participants in Experiment 2 followed the same procedure as those in Experiment 1. Only the personality vignettes differed (see below).

**Vignettes.**    As in Experiment 1, participants were randomly presented with a single vignette purportedly describing the trustee paired with a video clip displaying their facial expression. These modified vignettes aimed to represent a qualitatively different phenotype with a level of severity and psychosocial functioning comparable to those used in Experiment 1. The vignette portraying borderline personality disorder read as follows:

> Kaci is a 25-year-old woman who has had several intense and stormy relationships within the past year. They all seem to follow the same pattern. Kaci shifts between thinking that her partner is the best person in the world to quickly hating everything about him. During these relationships, she often becomes intensely angry and has even gotten into physical fights with her partner. These episodes are followed by a strong fear that the person will leave without warning, never to return. After the relationships end, Kaci often feels profoundly empty and lonely. And, as a result, doesn't have a strong sense of who she is when she's not with someone. When things are most difficult, she often seems "vacant" to others. She also has superficial cuts along her forearms and wrists.

The vignette portraying antisocial personality disorder read as follows:

> Tammy is a 25-year-old woman who has a long-standing pattern of disregard for the rights of others. She was recently arrested for attempting to con an acquaintance out of several hundred dollars, by pretending to be a debt collector. When she was put on probation for the crime, she rationalized her behavior by stating that her acquaintance deserved it for being so gullible. She showed no remorse. This is not the first time that Tammy has been in trouble with the law. She has been arrested in the past for physical assault and theft. Much of the time these are the result of her impulsive and reckless behaviors. Tammy has not shown a consistent ability to sustain work and has several unpaid bills and debts to her name.

The control vignette was identical to that used in Experiment 1.

## Results

**Preliminary results.**    As a manipulation check, we first conducted a 3 (personality) x 2 (expression) analysis of variance (ANOVA) to examine the effects on participants' ratings of the confederate's perceived happiness. There was a significant main effect for personality, $F(2, 277) = 3.91$, $p = .021$, $\eta2 = .027$. Post hoc LSD tests revealed that participants who read the control vignette rated the person depicted in the vignette as significantly more happy than participants who read the borderline vignette, mean difference = .57 ($SE = .20$); $p = .006$, 95% CI = [.17, .97]. Participants who read the control vignette also rated the person depicted in the vignette as significantly more happy than participants who read the antisocial vignettes, mean difference = .43 ($SE = .20$); $p = .035$, 95% CI = [.03, .84]. However, there was no significant different between participants who read the borderline vignette and those who read the antisocial vignette, mean difference = -.13 ($SE = .21$), $p = .516$, 95% CI = [-.54, .27].

There was also a significant main effect for facial expression, $F(1, 277) = 430.38$, $p < .001$, $\eta2 = .608$. Averaged across personality vignettes, perceived happiness ratings were higher among those who viewed the smile ($M = 5.42$, $SD = 1.40$), compared to those who viewed the

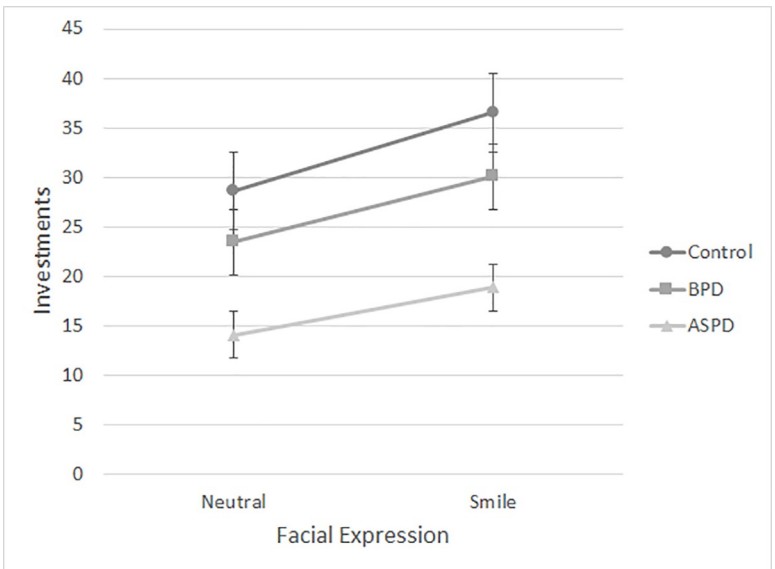

**Fig 2. Money invested (in cents) by vignette and facial expression, Experiment 2.** BPD = borderline personality disorder. ASPD = antisocial personality disorder.

neutral expression ($M$ = 1.95, $SD$ = 1.43). There was no significant personality by expression interaction, $F$(2, 277) = 0.192, $p$ = .825, η2 = .001.

**Primary results.**   The data of interest are shown in Fig 2. A 3 (personality) x 2 (expression) ANOVA was conducted to examine the effects on participants' investments. There was a significant main effect of personality, $F$(2, 277) = 17.15, $p$ < .001, η2 = .110. Post hoc LSD tests revealed that participants who read the control vignette invested significantly more (32.57 cents, $SD$ = 17.94) than participants who read the borderline vignette (26.71 cents, $SD$ = 20.31), mean difference = 5.87; $SE$ = 2.76; $p$ = .034, 95% CI for the mean difference = [0.44, 11.29]). They also invested significantly more (mean difference = 16.15; $SE$ = 2.78; $p$ < .001, 95% CI for the mean difference = [10.68, 21.62]) after reading the control vignette than they did after reading the antisocial vignette where the mean investment was only 16.42 cents ($SD$ = 19.37). There was also a significant mean difference of 10.28 ($SE$ = 2.79) between the investments made by participants who read the borderline vignette in comparison to those who read the antisocial vignette, $p$ < .001, 95% (CI) for the mean difference = [4.80, 15.76].

There was also a significant main effect of expression, $F$(1, 277) = 8.06, $p$ = .005, η2 = .028. Averaged across personality vignettes, investments were higher among those who viewed the smile ($M$ = 28.67, $SD$ = 19.59) compared to those who viewed the neutral expression ($M$ = 22.10, $SD$ = 20.49). There was no significant personality by facial expression interaction, $F$(2, 277) = 0.167, $p$ = .846, η2 = .001.

## General discussion

In two experiments, we examined the effects of information about a partners' personality and facial expression on prosocial behavior in a trust game. Results of both experiments supported our hypotheses when comparing participants' behavior in response to the control and borderline vignettes. When displaying either a neutral expression or a smile, participants made smaller investments towards those described as having borderline personality disorder in comparison to those described as having no personality pathology. In addition, if the person they believed they were interacting with was described as having either borderline personality

pathology or no personality pathology, participants made larger investments to confederates displaying a smile versus a neutral expression. This is consistent with previous research suggesting that knowledge of a partner's borderline pathology decreases prosocial behavior [1, 2] whereas the presence of smiles increases prosocial behavior [8–10].

Interestingly, results were not entirely consistent with our hypotheses when comparing participants' behavior in response to the control and antisocial vignettes. In Experiment 1, we found that the effects of facial expression varied depending on whether or not the vignette described antisocial personality traits. More specifically, if their partner was described as having antisocial traits, the presence of a smile *decreased* how much participants were willing to invest. In contrast, if the partner was described as having no antisocial traits, a smile *increased* participant's investments. This is consistent with previous research suggesting that knowledge of a partner's antisocial personality pathology decreases prosocial behavior [1, 2]. However, it specifies a caveat to previous research demonstrating that smiles increase prosocial behavior. These results suggest that this may only be the case when specific information regarding personality pathology does not explicitly counter judgements of trustworthiness. To our knowledge, this is the first empirical evidence demonstrating such an effect of felt smiles.

This effect, however, was not replicated in Experiment 2. Here, the results showed only main effects for personality and facial expression. That is, participants made smaller investments towards those described as having antisocial personality disorder in comparison to those described as having no pathology across facial expressions. Furthermore, participants made larger investments towards those displaying a smile in comparison to a neutral expression across personality vignettes. It is possible that the decrease in trust associated with smiling in the context of antisocial traits in Experiment 1 represents a Type 1 error. Another possibility is that differences in the findings from Experiment 1 and Experiment 2 are a result of differences in the features of antisocial personality disorder that are described in the two vignettes. In other words, heterogeneity in antisocial pathology may play a key role in determining whether smiling facilitates or diminishes trust. Antisocial phenotypes containing information about conning others (as was the case in Experiment 1) may not only diminish trust but also make people inclined to view smiling in this context with some degree of suspicion. In such contexts, the degree of suspicion can result in a decrease in prosocial behavior above and beyond those seen in borderline pathology. Future research should explore this possibility by systematically varying the specific diagnostic features present in personality vignettes.

These effects may impact the interpersonal functioning of individuals with personality pathology. Smiling may be an important component for individuals with borderline pathology who aim to be perceived as trustworthy and elicit prosocial behavior. Within the context of borderline pathology, a smile is perceived as a signal of increased trustworthiness. However, this same component could have either positive or negative effects on psychosocial functioning for individuals with antisocial pathology. Within the context of antisocial pathology, a smile can be perceived as a signal of increased trustworthiness or an intent to behave dishonestly. Taken together, these findings speak to the importance of both expressive characteristics and specific diagnostic features in the study of interpersonal functioning in personality pathology. Each plays an important role in affecting social interactions that may nuance future research studies and psychosocial interventions.

These results must be interpreted within the context of limitations. Although we believe our vignettes were successful in conveying personality pathology, they may not reflect real-world encounters characterized by incomplete information and where individuals may have the opportunity to actively manage impressions. That is, it is possible that individuals with borderline and/or antisocial personality pathology would present themselves in ways that elicit behavioral responses that differ from those in our findings. Similarly, we examined behavior in

a one-shot, as opposed to an iterated, game. As such, it remains to be seen how these behaviors might change over time in the context of more frequent interactions and experiences with a partner. Additionally, we did not collect any information on the personality pathology of the participants themselves. This would allow for the examination of the ways that individuals with specific personality pathology behave towards others with specific personality pathology. Finally, although we have no reason to believe otherwise, we did not confirm whether participants believed they were participating with a real person.

Our findings speak not only to the combined effects of knowledge regarding personality pathology and facial expression on prosocial behavior, but also demonstrate the promise held in integrating research in behavioral economics and psychological science. Future research integrating findings from each field can shed further light on the independent and combined effects of personality, emotion, signaling, and judgment on prosocial behavior. They might also lead to innovations within each field that yield further novel research findings.

## Author Contributions

**Conceptualization:** Lawrence Ian Reed, Ashley K. Meyer, Sara J. Okun, Cheryl K. Best, Jill M. Hooley.

**Data curation:** Lawrence Ian Reed.

**Formal analysis:** Lawrence Ian Reed.

**Investigation:** Lawrence Ian Reed.

**Methodology:** Lawrence Ian Reed.

**Project administration:** Lawrence Ian Reed.

**Software:** Lawrence Ian Reed.

**Writing – original draft:** Lawrence Ian Reed, Jill M. Hooley.

**Writing – review & editing:** Lawrence Ian Reed, Ashley K. Meyer, Sara J. Okun, Cheryl K. Best, Jill M. Hooley.

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
