## [Decision Letter · Decision Letter 0]

2 Apr 2020

PONE-D-20-06257

In Smiles We Trust? Smiling in the Context of Antisocial and Borderline Personality Pathology

PLOS ONE

Dear Dr. Reed,

Thank you for submitting your manuscript to PLOS ONE. After careful consideration, we feel that it has merit but does not fully meet PLOS ONE’s publication criteria as it currently stands. Therefore, we invite you to submit a revised version of the manuscript that addresses the points raised during the review process.

In the revision, attention should be given to lack of study rationale and hypothesis and relevant analyses, for example, Reviewer 1 suggested to use one 2x3 Anova. Please also clarify issues regarding the vignettes and how these were altered in Exp 2. Also consider comparing BPD to APD as suggested by Reviewer 2. Both reviewers provide suggestions of topics that should be addressed in the Discussion section.

We would appreciate receiving your revised manuscript by May 17 2020 11:59PM. To enhance the reproducibility of your results, we recommend that if applicable you deposit your laboratory protocols in protocols.io, where a protocol can be assigned its own identifier (DOI) such that it can be cited independently in the future. For instructions see: http://journals.plos.org/plosone/s/submission-guidelines#loc-laboratory-protocols

We look forward to receiving your revised manuscript.

Kind regards,

Robert Didden

Academic Editor

PLOS ONE

Journal Requirements:

Reviewers' comments:

Reviewer's Responses to Questions

**Comments to the Author**

1. Is the manuscript technically sound, and do the data support the conclusions?

Reviewer #1: Yes

Reviewer #2: Yes

2. Has the statistical analysis been performed appropriately and rigorously? 

Reviewer #1: Yes

Reviewer #2: Yes

3. Have the authors made all data underlying the findings in their manuscript fully available?

Reviewer #1: Yes

Reviewer #2: Yes

4. Is the manuscript presented in an intelligible fashion and written in standard English?

Reviewer #1: Yes

Reviewer #2: Yes

5. Review Comments to the Author

Reviewer #1: The authors conduct two experimental studies, using MTurk samples, of how much trust individuals have in others who display traits of borderline or antisocial PD vs. ostensibly non-impaired control others. They find main effects of personality pathology and of whether the ostensible targets are smiling, and in one case an interaction with theoretically interesting implications (although this does not replicate). Strengths of the manuscript include a straightforward design and an attempt to conceptually replicate an initial finding. I did have a few concerns to note:

1. The authors had seemingly no a priori hypotheses about the interaction of personality pathology and smiling on perceived trustworthiness. This is a puzzle, because they naturally test this interaction as a part of their analyses (and indeed interaction effects are one of the motivations for conducting a two-way ANOVA in the first place, as opposed to separate t-tests). I do not suggest that they invent a hypothesis now on a post hoc basis, just that it would have been preferable if they had considered this.

2. It would help to have more detail about how the borderline and antisocial vignettes were altered for experiment 2. The authors say only that each vignette “portray[s] another phenotype with several defining features of that personality disorder functioning outside of a hospital setting,” but the nature of the changes is not clear. Are fewer features of the disorder described? A lower severity? A qualitatively different phenotype? These details are important to help the reader interpret differences in results between the two experiments.

3. I would prefer it if the authors did one 2 x 3 ANOVA for each experiment rather than two 2 x 2 ANOVAs. Not only will this help the inferential statistics line up with the Figures, but this will also help them manage Type I error as well as permit a direct comparison of BPD and ASPD conditions.

4. A limitation of the study not noted by the authors was that the trust game was a one-shot proposition and not iterated. Thus, we don’t know how these interactions might play out in the long term.

5. Do the authors have any idea whether the deception that the targets were real individuals was credible to the participants? This would seem to be an important thing to check for, perhaps during debriefing. If they know this, they should report it. If they don’t, this is a major limitation of the study.

6. Error bars should be added to each figure.

Reviewer #2: The manuscript entitled “In smiles we trust? Smiling in the context of antisocial and borderline personality pathology” examined the extent to which smiles and personality pathology individually and concurrently contribute to trust in the trust game. The study benefitted from a number of strengths including two large samples with similar but slightly different experimental designs. Below I will address my concerns.

Introduction

-I suggest the authors provide a concrete rationale for the examination of borderline personality disorder and antisocial personality disorder. Given that the studies mentioned by Reed and colleagues (2018) suggest that behavior changes across a number of personality disorders, could the authors more concretely state the reasons behind the examination of borderline and antisocial alone.

- I wonder if the authors could flesh out why examining the associations between personality pathology, smiling, and trust is an important area of inquiry. I think the introduction nicely lays out why there is theoretical reason to believe that there may be a discrepancy between personality pathology and trust, but could the authors also address how this will impact the study of personality disorders? For example, what impact will this have on how we think about social interactions for personality disorders? Diagnostic features? Etc.

-Perhaps components of the paragraphs between experiment 1 and experiment 2 should be moved to the introduction.

-did the authors have any hypotheses regarding whether smiling + personality pathology versus smiling + no personality pathology would lead to differences in trustworthiness? Or was this part of the study exploratory? Either way, this should be stated.

-do the authors have a rationale for why they chose the trust game as opposed to another gaming paradigm? Could they emphasize why they made this choice? Does it capture a specific social preference beyond other gaming paradigms? Has it been shown to more strongly relate to trustworthiness than others?

Method

-were participants led to believe their take-home pay was impacted by offer exchanges? Or were they told ahead of time that it would have no impact on their pay for the experiment?

Results

-All results include comparison of borderline personality disorder to no personality disorder and antisocial personality disorder to no personality disorder. Results comparing borderline personality disorder to antisocial personality disorder should also be conducted. This would have important implications for diagnosis, as the authors suggest in the discussion.

Discussion

-Given that a main component of the study was to examine the effect of smiling with and without a believed personality disorder, the authors should address what it means to smile in the context of personality disorders. For example, if individuals with borderline personality pathology strive to be viewed as more trustworthy, smiling may be an important component. The same might not be true for individuals with antisocial personality pathology.

-the point of heterogeneity within personality disorder diagnosis is an important one and should be fleshed out. This point would also be stronger if the authors could claim that the antisocial traits in experiment one contribute to a lack of trustworthiness above and beyond the borderline personality disorder features.

-in a similar vein, including this comparison would allow the researchers to discern whether the instability (as described in borderline personality disorder vignette) is viewed as entirely different than deceiving others (as described in the antisocial vignette) and what this means about trust.

6. PLOS authors have the option to publish the peer review history of their article (what does this mean?). If published, this will include your full peer review and any attached files.

Reviewer #1: Yes: William D Ellison

Reviewer #2: No

---

## [Author Response · Author response to Decision Letter 0]

11 May 2020

May 11, 2020

Dr. Robert Didden

Academic Editor

PLOS ONE

Dear Dr. Didden,

Thank you for the opportunity to revise our manuscript, “In Smiles We Trust? Smiling in the context of antisocial and borderline personality pathology” (PONE-D-20-06257). Both reviewers have raised important points. We are grateful for their thoughtful feedback. 

We have revised the manuscript incorporating the suggestions of both reviewers to the best of our ability. We believe the manuscript is now considerably improved as a result. Specific details of the revisions are detailed below.

We thank you again for your helpful comments and suggestions. We hope the changes that have been made meet with your approval. We look forward to hearing from you in due course.

Lawrence Ian Reed, Ph.D.

Ashley K. Meyer

Sara J. Okun

Cheryl K. Best

Jill M. Hooley

 

Reviewer #1

1. The authors had seemingly no a priori hypotheses about the interaction of personality pathology and smiling on perceived trustworthiness. This is a puzzle, because they naturally test this interaction as a part of their analyses (and indeed interaction effects are one of the motivations for conducting a two-way ANOVA in the first place, as opposed to separate t-tests). I do not suggest that they invent a hypothesis now on a post hoc basis, just that it would have been preferable if they had considered this.

Our examination of the combined effects of personality and expression were exploratory. We have now explicitly stated this.

New text (p. 6):

“Finally, we examined the combined effects of personality and smiles to explore potential interactions.”

New text (p. 14):

“Also, as in Experiment 1, we examined the combined effects of personality and smiles to explore potential interactions.”

2. It would help to have more detail about how the borderline and antisocial vignettes were altered for experiment 2. The authors say only that each vignette “portray[s] another phenotype with several defining features of that personality disorder functioning outside of a hospital setting,” but the nature of the changes is not clear. Are fewer features of the disorder described? A lower severity? A qualitatively different phenotype? These details are important to help the reader interpret differences in results between the two experiments.

We agree that these differences are critical in the motivation and results for Experiment 2. In response, we have added prose describing the aim of these vignettes.

New text (p. 15):

“These modified vignettes aimed to represent a qualitatively different phenotype with a level of severity and psychosocial functioning comparable to those used in Experiment 1.”

3. I would prefer it if the authors did one 2 x 3 ANOVA for each experiment rather than two 2 x 2 ANOVAs. Not only will this help the inferential statistics line up with the Figures, but this will also help them manage Type I error as well as permit a direct comparison of BPD and ASPD conditions.

We agree with Reviewer #1 and Reviewer #2 (see Reviewer #2’s 7th point below) regarding the analyses. In response, we have now conducted a 3 (personality) by 2 (expression) ANOVA as the primary analyses in Experiments 1 and 2. This is now more consistent with the figures and allows for the comparison of investments among those who read the borderline and antisocial vignettes.

Old text:

“A 2 (personality) x 2 (expression) analysis of variance (ANOVA) comparing those who read the neutral and borderline vignettes revealed a significant main effect for personality disorder F(1, 171) = 10.66, p = .001, η2 = .059, indicating that participants transferred less money to the confederate after reading the borderline vignette compared to the control vignette. There was also a significant main effect for facial expression F(1, 171) = 3.21, p = .016, η2 = .033 indicating that participants gave more money after viewing the smiling clip than they did after viewing the neutral expression clip. There was no significant interaction, F(1, 171) = 0.04, p = .843, η2 < .001.

Among those who read the antisocial personality disorder vignette, participants shared an average of 23.67 (SD = 21.37) if they viewed the neutral clip and 16.78 (SD = 19.07) if they viewed the smiling clip. A 2 (personality) x 2 (expression) ANOVA comparing those who read the neutral and antisocial vignettes revealed a significant main effect for personality disorder F(1, 171) = 21.25, p < .001, η2 = .111 indicating that participants gave less to the confederate after reading the antisocial vignette in comparison to the control vignette. There was no significant main effect for facial expression F(1, 171) = 0.04, p = .852, η2 < .001. However, these results were qualified by a significant interaction, F(1, 171) = 4.96, p = .027, η2 = .028, indicating that the effects of expression were dependent upon which vignette the participant read. More specifically, participants who viewed the smiling clip offered more money if they read the control vignette but offered less money if they read the antisocial vignette.”

New text (p. 12/13):

“A 3 (personality) x 2 (expression) ANOVA was conducted to examine the effects of personality and facial expression on participants’ investments. There was a significant main effect of personality, F(2, 256) = 11.67, p < .001, η2 = .084, but no significant main effect of expression, F(1, 256) = 0.73, p = .394, η2 = .003. These main effects were also qualified by a significant interaction, F(2, 256) = 3.82, p = .023, η2 = .029.

The significant personality by expression interaction effect was analyzed using a simple main effects analysis. Among participants who viewed the neutral expression, investments did not significantly differ across personality vignettes F(2, 256) = 2.843, p = .060, η2 = .022, although investments in partners depicted with features of borderline or antisocial personality were lower (21.44 cents; SD=17.66 and 23.66 cents; SD = 21.37 respectively) than they were for the control vignette (30.47 cents; SD= 18.65). However, if participants viewed the smile, investments differed significantly across the personality vignettes, F(2, 256) = 12.967, p < .001, η2 = .092. Post-hoc LSD tests revealed that, after having viewed a smile, participants who read the control vignette invested more (36.39 cents; SD=16.7) than participants who read the borderline vignette (28.30 cents; SD= 16.35), mean difference = 7.99, SE = 3.87; p = .040, 95% confidence interval (CI) for the mean difference = [0.37, 15.62]. They also invested significantly more (mean difference = 19.51; SE = 3.85; p < .001, 95% CI for the mean difference = [11.93, 27.30]) after reading the control vignette than they did after reading the antisocial vignette where the mean investment was only 16.78 cents (SD=19.07). There was also a significant mean difference of 11.52 (SE = 3.87) between the investments made by participants who read the borderline vignette in comparison to those who read the antisocial vignette, p = .003, 95% (CI) for the mean difference = [3.89, 19.15]. After having viewed a smile, participants gave less money to the person depicted in the antisocial vignette compared to the person depicted in the borderline vignette. The data of interest are presented in Figure 1.

Results from Experiment 1 suggest that both knowledge of a partner’s personality pathology and facial expression affect trust. This is consistent with previous behavioral work examining personality pathology (Reed, et al., 2018; Reed, et al., 2018) and smiles (Brown & Moore, 2002; Mehu, et al., 2007; Reed, et al., 2012; Scharlemann, et al., 2001) in economic games. In response to the borderline personality disorder vignette, these effects were independent. However, in response to the antisocial personality disorder vignette, the effects combined to create an antagonistic interaction.”

Old text:

“A 2 (personality) x 2 (expression) ANOVA comparing those who read the neutral and borderline vignettes revealed a significant main effect for personality disorder F(1, 191) = 5.55, p = .034, η2 = .024, indicating that participants shared less with the confederate after reading the borderline vignette in comparison to reading the control vignette. There was also a significant main effect for facial expression F(1, 191) = 7.07, p = .009, η2 = .036 indicating that participants shared more after viewing the smiling clip than they did after viewing the neutral expression. There was no significant interaction, F(1, 191) = .059, p = .808, η2 < .001.

A 2 (personality) x 2 (expression) ANOVA comparing those who read the neutral and antisocial vignettes revealed a significant main effect for personality disorder F(1, 188) = 35.73, p < .001, η2 = .163, indicating that participants shared less with the confederate after reading the antisocial vignette in comparison to reading the control vignette. There was also a significant main effect for facial expression F(1, 188) = 5.54, p = .020, η2 = .029 indicating that participants shared more after viewing the smiling clip in comparison to viewing the neutral expression. There was no significant interaction, F(1, 188) = .353, p = .553, η2 = .002.”

“The data of interest are shown in Figure 2. A 3 (personality) x 2 (expression) ANOVA was conducted to examine the effects on participants’ investments. There was a significant main effect of personality, F(2, 277) = 17.15, p < .001, η2 = .110. Post hoc LSD tests revealed that participants who read the control vignette invested significantly more (32.57 cents, SD = 17.94) than participants who read the borderline vignette (26.71 cents, SD = 20.31), mean difference = 5.87; SE = 2.76; p = .034, 95% CI for the mean difference = [0.44, 11.29]). They also invested significantly more (mean difference = 16.15; SE = 2.78; p < .001, 95% CI for the mean difference = [10.68, 21.62]) after reading the control vignette than they did after reading the antisocial vignette where the mean investment was only 16.42 cents (SD = 19.37). There was also a significant mean difference of 10.28 (SE = 2.79) between the investments made by participants who read the borderline vignette in comparison to those who read the antisocial vignette, p < .001, 95% (CI) for the mean difference = [4.80, 15.76].

Figure 2. Money invested (in cents) by vignette and facial expression, Experiment 2. BPD = borderline personality disorder. ASPD = antisocial personality disorder. 

There was also a significant main effect of expression, F(1, 277) = 8.06, p = .005, η2 = .028. Averaged across personality vignettes, investments were higher among those who viewed the smile (M = 28.67, SD = 19.59) compared to those who viewed the neutral expression (M = 22.10, SD = 20.49). There was no significant personality by facial expression interaction, F(2, 277) = 0.167, p = .846, η2 = .001.”

Considering the changes to our primary analyses in Experiments 1 and 2, we have also changed our preliminary analyses. Previously, and as we had for our primary analyses, we conducted separate 2 (expression) by 2 (personality) ANOVAs. One comparing those who read the control vignette with those who read the borderline vignette and a second comparing those who read the control vignette with those who read the antisocial vignette. Here, we have conducted a single 2 (expression) by 3 (personality) ANOVA in each experiment. We hope that you and the reviewers are amendable to this change.

Old text:

“As a manipulation check, we first compared participants’ ratings of the confederate trustee’s happiness between those who read the control vignette and those who read the borderline vignette. Among those who read the control vignette, participants’ happiness ratings were an average of 2.41 (SD = 1.31) if they viewed the neutral clip and 5.64 (SD = 1.17) if they viewed the smiling clip. Among those who read the borderline vignette, participants’ happiness ratings were an average of 2.05 (SD = 1.45) if they viewed the neutral clip and 5.55 (SD = 1.37) if they viewed the smiling clip. A 2 (personality) x 2 (expression) analysis of variance ANOVA comparing those who read the neutral and borderline vignettes revealed no significant main effect for personality disorder, F(1, 171) = 1.38, p = .242. However, there was a significant main effect for expression, F(1, 171) = 280.35, p < .001, such that participants rated the smiling person as being happier. Personality did not significantly interact with expression, F(1, 171) = 0.46, p = .497.

We then compared participants’ ratings of the confederate trustee’s happiness between those who read the control vignette and those who read the antisocial vignette. For those who read the antisocial vignette, participants’ happiness ratings averaged 1.81 (SD = 1.06) if they viewed the neutral clip and 5.04 (SD = 1.72) if they viewed the smiling clip. A 2 (personality) x 2 (expression) analysis of variance ANOVA comparing those who read the control and antisocial vignettes revealed significant main effects for personality disorder, F(1, 171) = 8.84, p = .003, and expression, F(1, 171) = 252.30, p < .001; the person depicted in the control vignette was rated as being happier than the person in the antisocial vignette and the smiling person was rated as being happier than the person with a neutral expression. Personality did not significantly interact with expression, F(1, 171) < 0.01, p = .982.”

New text (p. 11/12):

“As a manipulation check, we first conducted a 3 (personality) x 2 (expression) analysis of variance (ANOVA) to examine the effects on participants’ ratings of the confederate’s perceived happiness. There was a significant main effect for personality, F(2, 256) = 4.34, p = .014, η2 = .033. Post hoc LSD tests revealed no significant difference between participants who read the control vignette and those who read the borderline vignette, mean difference = .25 (SE = .21), p = .223, 95% confidence interval (CI) = [-.15, .66]. However, participants who read the antisocial vignette rated the person depicted in the vignette as significantly less happy than participants who read the control vignette (mean difference = .59; SE = .21; p = .005, 95% CI = [.18, .99)]. Ratings of perceived happiness did not differ significantly between participants who read the borderline vignette and those who read the antisocial vignette, mean difference = .33 (SE = .21), p = .109, 95% CI = [-.07, .74].

There was also a significant main effect for expression, F(1, 256) = 386.41, p < .001, η2 = .601. Averaged across personality vignettes, perceived happiness ratings were higher among those who viewed the smile (M = 5.41, SD = 1.45) compared to those who viewed the neutral expression (M = 2.09, SE = 1.30). There was no significant personality by expression interaction, F(1, 256) = .0.28, p = .755, η2 = .002.”

Old text:

“As a manipulation check, we first compared participants’ ratings of the confederate trustee’s happiness when they were exposed to the control vignette and the borderline vignette. Among those who read the control vignette, participants’ happiness ratings were an average of 2.21 (SD = 1.38) if they viewed the neutral clip and 5.79 (SD = 1.29) if they viewed the smiling clip. Among those who read the borderline vignette, participants’ happiness ratings were an average of 1.71 (SD = 1.09) if they viewed the neutral clip and 5.19 (SD = 1.42) if they viewed the smiling clip. A 2 (personality) x 2 (expression) ANOVA comparing those who read the neutral and borderline vignettes revealed a significant main effect for personality disorder, F(1, 191) = 8.522, p = .004. There was also a significant main effect for expression, F(1, 191) = 351.585, p < .001, such that participants rated the smiling person as being happier. Personality did not significantly interact with expression, F(1, 191) = .071, p = .791 in the prediction of confederate happiness. 

We then compared participants’ ratings of the confederate trustee’s happiness between those who read the control vignette and those who read the antisocial vignette. For those who read the antisocial vignette, participants’ happiness ratings averaged 1.94 (SD = 1.72) if they viewed the neutral clip and 5.27 (SD = 1.45) if they viewed the smiling clip. A 2 (personality) x 2 (expression) ANOVA comparing those who read the control and antisocial vignettes revealed no significant main effect for personality disorder, F(1, 188) = 3.45, p = .065. However, there was a significant main effect for expression, F(1, 188) = 259.85, p < .001; the person depicted in the control vignette was rated as being happier than the person in the antisocial vignette and the smiling person was rated as being happier than the person with a neutral expression. Personality did not significantly interact with expression, F(1, 188) = .348, p = .556.”

New text (p. 15/16):

“As a manipulation check, we first conducted a 3 (personality) x 2 (expression) analysis of variance (ANOVA) to examine the effects on participants’ ratings of the confederate’s perceived happiness. There was a significant main effect for personality, F(2, 277) = 3.91, p = .021, η2 = .027. Post hoc LSD tests revealed that participants who read the control vignette rated the person depicted in the vignette as significantly more happy than participants who read the borderline vignette, mean difference = .57 (SE = .20); p = .006, 95% CI = [.17, .97]. Participants who read the control vignette also rated the person depicted in the vignette as significantly more happy than participants who read the antisocial vignettes, mean difference = .43 (SE = .20); p = .035, 95% CI = [.03, .84]. However, there was no significant different between participants who read the borderline vignette and those who read the antisocial vignette, mean difference = -.13 (SE = .21), p = .516, 95% CI = [-.54, .27].

There was also a significant main effect for facial expression, F(1, 277) = 430.38, p < .001, η2 = .608. Averaged across personality vignettes, perceived happiness ratings were higher among those who viewed the smile (M = 5.42, SD = 1.40), compared to those who viewed the neutral expression (M = 1.95, SD = 1.43). There was no significant personality by expression interaction, F(2, 277) = 0.192, p = .825, η2 = .001.”

4. A limitation of the study not noted by the authors was that the trust game was a one-shot proposition and not iterated. Thus, we don’t know how these interactions might play out in the long term.

We agree with the reviewers point and have added this limitation to our discussion.

New text (p. 19):

“Similarly, we examined behavior in a one-shot, as opposed to an iterated game. As such, it remains to be seen how these behaviors might change over time.”

5. Do the authors have any idea whether the deception that the targets were real individuals was credible to the participants? This would seem to be an important thing to check for, perhaps during debriefing. If they know this, they should report it. If they don’t, this is a major limitation of the study.

We asked participants for feedback after the debriefing and did not receive any comments that suggested they were doubtful they were participating with a real person. However, we did not confirm this and have thus listed it as a limitation in the discussion section.

New text (p. 19):

“Finally, otherwise we have no reason to believe they did not, we did not confirm whether participants believed they were participating with a real person.”

6. Error bars should be added to each figure.

We apologize for the oversight. We thank the reviewer for this comment and have added error bars to the revised figures.

Reviewer #2

1. I suggest the authors provide a concrete rationale for the examination of borderline personality disorder and antisocial personality disorder. Given that the studies mentioned by Reed and colleagues (2018) suggest that behavior changes across a number of personality disorders, could the authors more concretely state the reasons behind the examination of borderline and antisocial alone.

We chose to investigate borderline and antisocial personality pathology because they have shown among the strongest effects in our previous studies on cooperation and bargaining. We have added prose to specify this.

New text (p. 5):

“The presence of borderline and antisocial personality pathology have among the strongest effects on cooperation and bargaining (Reed, et al., 2018; Reed, et al., 2018).”

2. I wonder if the authors could flesh out why examining the associations between personality pathology, smiling, and trust is an important area of inquiry. I think the introduction nicely lays out why there is theoretical reason to believe that there may be a discrepancy between personality pathology and trust, but could the authors also address how this will impact the study of personality disorders? For example, what impact will this have on how we think about social interactions for personality disorders? Diagnostic features? Etc.

In response, we have added prose to the discussion further highlighting the effects of specific diagnostic features and expressive characteristics in future studies and psychosocial interventions.

New text (p. 19):

“Taken together, these findings speak to the importance of both expressive characteristics and specific diagnostic features in the study of interpersonal functioning in personality pathology. Each plays an important role in affecting social interactions that may nuance future research studies and psychosocial interventions.”

3. Perhaps components of the paragraphs between experiment 1 and experiment 2 should be moved to the introduction. 

In response, we have moved the following paragraph (previously located between Experiment 1 and Experiment 2) to the final paragraph in the introduction.

Moved text (p. 5):

“Both borderline and antisocial personality disorders are defined polythetically (i.e. a person must meet a minimum number of diagnostic criteria to warrant a diagnosis) (American Psychiatric Association, 2013). As such, there exists a great deal of potential heterogeneity among individuals who are diagnosed with either form of personality disorder. To examine the potential effects of varying the presentation of personality pathology, we modified the personality disorder vignettes in a second experiment to portray another phenotype with several defining features of each personality disorder functioning in a community setting.

Based on previous research, we hypothesized that confederate trustees whose descriptions contained features of either borderline or antisocial personality disorder would be judged as less trustworthy and given smaller investments from participant investors in comparison to those described as having no pathology. We also hypothesized that confederates who smiled, regardless of the presence or absence of a personality disorder, would be judged as more trustworthy and given larger investments in comparison to those who displayed a neutral expression. Finally, we examined the combined effects of personality and smiles to explore potential interactions.”

We have also moved the following paragraph (previously in the beginning of Experiment 2) to the end of Experiment 1.

Moved text (p. 13):

“Results from Experiment 1 suggest that both knowledge of a partner’s personality pathology and facial expression affect trust. This is consistent with previous behavioral work examining personality pathology (Reed, et al., 2018; Reed, et al., 2018) and smiles (Brown & Moore, 2002; Mehu, et al., 2007; Reed, et al., 2012; Scharlemann, et al., 2001) in economic games. In response to the borderline personality disorder vignette, these effects were independent. However, in response to the antisocial personality disorder vignette, the effects combined to create an antagonistic interaction.”

4. Did the authors have any hypotheses regarding whether smiling + personality pathology versus smiling + no personality pathology would lead to differences in trustworthiness? Or was this part of the study exploratory? Either way, this should be stated.

Please see the above response to Reviewer #1’s first comment above.

5. Do the authors have a rationale for why they chose the trust game as opposed to another gaming paradigm? Could they emphasize why they made this choice? Does it capture a specific social preference beyond other gaming paradigms? Has it been shown to more strongly relate to trustworthiness than others?

In response, we have added prose stating that we chose the trust game because it is a valid measure of trust that would yield data that is complementary to our previous works on personality pathology, facial expression, cooperation, and bargaining. We have also added two citations demonstrating that the trust game validly measures trust (as opposed to altruism).

New text (p. 5):

“We aimed to examine trust to further extend our previous research on personality pathology, facial expression, cooperation, and bargaining. Previous work has demonstrated that smiles increase prosocial behavior (Reed, et al., 2012). The presence of borderline and antisocial personality pathology also influences cooperation and bargaining behavior in those who play economic games with them (Reed, et al., 2018; Reed, et al., 2018). Here, we examined participants’ behavior in a one-shot Trust game towards confederates who varied in both personality pathology (using vignettes describing borderline personality disorder, antisocial personality disorder, or no pathology) and facial expression (displaying either a neutral expression or a smile). We chose the one-shot trust game (Berg, et al., 1995) as it has been found to be a valid measure of trust (Brulhart & Usunier, 2012; Cox, 2004).”

6. Were participants led to believe their take-home pay was impacted by offer exchanges? Or were they told ahead of time that it would have no impact on their pay for the experiment?

Participants were told that the monetary payoffs they earned in the game would be paid to them in MTurk bonus payments before engaging in the task. We have now clarified the prose to reflect this.

Old text:

“Participants were paid $2.00 for completing the study. Importantly, participants were told that the monetary payoffs they earned in the game would be paid to them (via MTurk bonus payments).”

New text (p. 11):

“Participants were paid $2.00 for completing the study. Importantly, participants were told that the monetary payoffs they earned in the game would be paid to them (via MTurk bonus payments) before engaging in the task.”

7. All results include comparison of borderline personality disorder to no personality disorder and antisocial personality disorder to no personality disorder. Results comparing borderline personality disorder to antisocial personality disorder should also be conducted. This would have important implications for diagnosis, as the authors suggest in the discussion.

Please see the response to Reviewer #1’s 3rd point above.

8. Given that a main component of the study was to examine the effect of smiling with and without a believed personality disorder, the authors should address what it means to smile in the context of personality disorders. For example, if individuals with borderline personality pathology strive to be viewed as more trustworthy, smiling may be an important component. The same might not be true for individuals with antisocial personality pathology.

We agree. In response, we have added the following prose to the discussion.

New text (p. 18/19):

“These effects may impact the interpersonal functioning of individuals with personality pathology. Smiling may be an important component for individuals with borderline pathology who aim to be perceived as trustworthy and elicit prosocial behavior. Within the context of borderline pathology, a smile is perceived as a signal of increased trustworthiness. However, this same component could have either positive or negative effects on psychosocial functioning for individuals with antisocial pathology. Within the context of antisocial pathology, a smile can be perceived as a signal of increased trustworthiness or an intent to behave dishonestly.”

9. The point of heterogeneity within personality disorder diagnosis is an important one and should be fleshed out. This point would also be stronger if the authors could claim that the antisocial traits in experiment one contribute to a lack of trustworthiness above and beyond the borderline personality disorder features.

We agree that this is a crucial point. We have added a line of prose that expands upon the discussion of heterogeneity and alludes to the new findings from Experiment 1.

New text (p. 18):

“In such contexts, the degree of suspicion can result in a decrease in prosocial behavior above and beyond those seen in borderline pathology.”

10. In a similar vein, including this comparison would allow the researchers to discern whether the instability (as described in borderline personality disorder vignette) is viewed as entirely different than deceiving others (as described in the antisocial vignette) and what this means about trust.

We have included this comparison within the results of Experiment 1. Please see the response to Reviewer #1’s 3rd point above.

---

## [Decision Letter · Decision Letter 1]

26 May 2020

PONE-D-20-06257R1

In Smiles We Trust? Smiling in the Context of Antisocial and Borderline Personality Pathology

PLOS ONE

Dear Dr. Reed,

Thank you for submitting your manuscript to PLOS ONE. The revision has resulted in a much improved paper. I would be happy to accept your paper pending one very minor correction, that is adding error bars to the figures. In the response letter you responded that you have added error bars, but the reviewer (as well as myself) cannot see them in the uploaded file. See comment 6 from Reviewer #1. I would like to invite you to revise the paper one more time adding bars to the figures and submit.

We look forward to receiving your revised manuscript.

Kind regards,

Robert Didden

Academic Editor

PLOS ONE

Reviewers' comments:

Reviewer's Responses to Questions

**Comments to the Author**

1. If the authors have adequately addressed your comments raised in a previous round of review and you feel that this manuscript is now acceptable for publication, you may indicate that here to bypass the “Comments to the Author” section, enter your conflict of interest statement in the “Confidential to Editor” section, and submit your "Accept" recommendation.

Reviewer #1: (No Response)

Reviewer #2: All comments have been addressed

2. Is the manuscript technically sound, and do the data support the conclusions?

Reviewer #1: Yes

Reviewer #2: Yes

3. Has the statistical analysis been performed appropriately and rigorously? 

Reviewer #1: Yes

Reviewer #2: Yes

4. Have the authors made all data underlying the findings in their manuscript fully available?

Reviewer #1: Yes

Reviewer #2: Yes

5. Is the manuscript presented in an intelligible fashion and written in standard English?

Reviewer #1: Yes

Reviewer #2: Yes

6. Review Comments to the Author

Reviewer #1: The authors have responded quite thoroughly to my earlier comments and suggestions. It does seem as though some of the edited sections are different between the revised manuscript and the cover letter, but it appears that the manuscript contains the more up-to-date or authoritative revisions (which again are acceptable).

My only remaining critique is that error bars still seem to be missing in the figures supplied with the revised manuscript.

Reviewer #2: (No Response)

7. PLOS authors have the option to publish the peer review history of their article (what does this mean?). If published, this will include your full peer review and any attached files.

Reviewer #1: Yes: William D Ellison

Reviewer #2: No

---

## [Author Response · Author response to Decision Letter 1]

26 May 2020

May 26, 2020

Dr. Robert Didden

Academic Editor

PLOS ONE

Dear Dr. Didden,

We are delighted to hear that the revisions to our manuscript, “In Smiles We Trust? Smiling in the context of antisocial and borderline personality pathology” (PONE-D-20-06257) improved the paper. We have added the revised figures to include error bars to the current revision.

We thank you again for your helpful comments and suggestions and look forward to hearing from you in due course.

Lawrence Ian Reed, Ph.D.

Ashley K. Meyer

Sara J. Okun

Cheryl K. Best

Jill M. Hooley

---

## [Decision Letter · Decision Letter 2]

29 May 2020

In Smiles We Trust? Smiling in the Context of Antisocial and Borderline Personality Pathology

PONE-D-20-06257R2

Dear Dr. Reed,

We are pleased to inform you that your manuscript has been judged scientifically suitable for publication and will be formally accepted for publication once it complies with all outstanding technical requirements.

With kind regards,

Robert Didden

Academic Editor

PLOS ONE

Additional Editor Comments (optional):

Reviewers' comments:

Reviewer's Responses to Questions

**Comments to the Author**

1. If the authors have adequately addressed your comments raised in a previous round of review and you feel that this manuscript is now acceptable for publication, you may indicate that here to bypass the “Comments to the Author” section, enter your conflict of interest statement in the “Confidential to Editor” section, and submit your "Accept" recommendation.

Reviewer #1: All comments have been addressed

2. Is the manuscript technically sound, and do the data support the conclusions?

Reviewer #1: (No Response)

3. Has the statistical analysis been performed appropriately and rigorously? 

Reviewer #1: (No Response)

4. Have the authors made all data underlying the findings in their manuscript fully available?

Reviewer #1: (No Response)

5. Is the manuscript presented in an intelligible fashion and written in standard English?

Reviewer #1: (No Response)

6. Review Comments to the Author

Reviewer #1: (No Response)

7. PLOS authors have the option to publish the peer review history of their article (what does this mean?). If published, this will include your full peer review and any attached files.

Reviewer #1: Yes: William D Ellison

---

## [Editor Report · Acceptance letter]

15 Jun 2020

PONE-D-20-06257R2 

In Smiles We Trust? Smiling in the Context of Antisocial and Borderline Personality Pathology 

Dear Dr. Reed:

I'm pleased to inform you that your manuscript has been deemed suitable for publication in PLOS ONE. Congratulations! Your manuscript is now with our production department. 

Kind regards, 

on behalf of

Professor Robert Didden 

Academic Editor

PLOS ONE